# Proxy-Anchor and EVT-Driven Continual Learning Method for Generalized Category Discovery

**Alireza Fathalizadeh**                                              *alireza.fathalizadeh@unb.ca*
*Faculty of Computer Science*
*University of New Brunswick*

**Roozbeh Razavi-Far**                                              *roozbeh.razavi-far@unb.ca*
*Faculty of Computer Science*
*University of New Brunswick*

**Reviewed on OpenReview:** *https://openreview.net/forum?id=P3Qe9yJRvf*

## Abstract

Continual generalized category discovery has been introduced and studied in the literature as a method that aims to continuously discover and learn novel categories in incoming data batches while avoiding catastrophic forgetting of previously learned categories. A key component in addressing this challenge is the model's ability to separate novel samples, where Extreme Value Theory (EVT) has been effectively employed. In this work, we propose a novel method that integrates EVT with proxy anchors to define boundaries around proxies using a probability of inclusion function, enabling the rejection of unknown samples. Additionally, we introduce a novel EVT-based loss function to enhance the learned representation, achieving superior performance compared to other deep-metric learning methods in similar settings. Using the derived probability functions, novel samples are effectively separated from previously known categories. However, category discovery within these novel samples can sometimes overestimate the number of new categories. To mitigate this issue, we propose a novel EVT-based approach to reduce the model size and discard redundant proxies. We also incorporate a novel experience replay and knowledge distillation mechanisms during the continual learning stage to prevent catastrophic forgetting. Experimental results demonstrate that our proposed approach outperforms state-of-the-art methods in continual generalized category discovery scenarios. The implementation code is available at: `https://github.com/NumOne01/CATEGORIZER`

## 1 Introduction

Most traditional machine learning algorithms operate under the closed-world assumption, in which the training and test data are drawn from the same label and feature spaces and no new class is introduced to the model during the test phase. However, in a more practical scenario, the training data lacks complete knowledge of the world and unknown classes may emerge during the test phase. A crucial problem is that a model that operates under the closed set assumption usually makes high-confidence predictions for these novel samples. This is particularly problematic in critical systems like autonomous driving, where misclassification can cause serious harm, requiring the model to detect novel samples and integrate potential novel classes into the knowledge base of the model.

Humans have the ability to identify new entities and group them without prior knowledge, while also recognizing them upon subsequent encounters., e.g., one can see new types of bird species that they did not see before, but they can still group and categorize them once they see them and add this new information to their knowledge base without affecting what they know about other types of birds, i.e., no forgetting. This has inspired a scenario, called Continual Generalized Category Discovery (CGCD) Zhang et al. (2022); Wu

et al. (2023); Kim et al. (2023); Zhao & Mac Aodha (2023), in which a model is trained on an initially labeled dataset and after this initial stage, the model is only introduced to unlabeled data and is expected to detect and discover potential novel categories in the data and integrate them into the model without compromising the performance of previously learned tasks.

The problem of continual generalized category discovery can be decomposed into three subtasks: 1) **Novelty detection**: Detecting samples that do not belong to any previously learned and known categories Geng et al. (2020); Yang et al. (2024), 2) **Category discovery**: Identifying potential novel categories in an unlabeled dataset Han et al. (2021); Vaze et al. (2022), and 3) **Continual learning**: Integrating newly discovered categories into the model without catastrophic forgetting of previously learned categories Rolnick et al. (2019); Wang et al. (2024). Almost any method that solves each of these sub-tasks can be combined to handle the CGCD scenario, however, this might result in sub-optimal solutions as each method is trying to solve a different task, and balancing these competing objectives effectively is challenging Zhang et al. (2022). A unified framework tailored to this problem is then necessary to achieve a promising balance.

A crucial component of such a framework is the ability to clearly distinguish between known and unknown samples. To achieve this, we leverage Extreme Value Theory (EVT), which has demonstrated its effectiveness in addressing the open-set recognition problem Bendale & Boult (2016); Rudd et al. (2017); Geng et al. (2020). EVT is a statistical framework for modeling extreme deviations in data by analyzing the tail ends of the distribution Coles et al. (2001). In this context, EVT is used to model the margin distribution of each proxy, which is a representative of a class and learned as a part of the network parameters Kim et al. (2020), relative to other samples and define boundaries around proxies to reject unknown samples by developing a probability of inclusion function. Building on this, we propose a novel loss, called *evt* loss, which is derived from the EVT analysis of each proxy. In addition to preparing the model for unknown rejection, this loss improves the learned representation.

In the context of novel class discovery, existing methods often rely on clustering techniques, which tend to overestimate the number of classes. We mitigate this issue by utilizing EVT to reduce the number of estimated novel categories by discarding redundant ones, leading to improved discovery of new classes, while minimizing the forgetting of previously learned ones. To avoid catastrophic forgetting of previously learned data, we employ commonly used methods of knowledge distillation Li & Hoiem (2017); Hou et al. (2019) and experience replay Chaudhry et al. (2019); Rolnick et al. (2019); Lin et al. (2023); Zheng et al. (2024). During experience replay, we employ an EVT-driven distribution to sample features, which outperforms the Gaussian distribution commonly used in related work. In summary, our contribution can be summarized as follows:

- We propose a novel approach, called proxy-anchor and EVT-driven continual-learning method for generalized category discovery (CATEGORIZER). Extensive evaluations on multiple datasets demonstrate that our proposed method outperforms state-of-the-art approaches in the CGCD setting.

- We introduce a novel loss function called *evt* loss, which is derived from proxy anchors Kim et al. (2020) and extreme value theory. This loss outperforms deep metric learning methods used in similar methods.

- We propose a novel approach to mitigate over-estimating the number of novel categories in the discovery phase. This has been done by means of extreme value theory, which boosts the performance of the model in discovering categories as well as minimizing the forgetting of previously learned categories.

- We propose an EVT-driven Weibull-based feature replay strategy for continual learning and show that it consistently outperforms Gaussian-based approaches in prior work.

## 2 Related Works

**Novelty detection** aims to identify samples from novel classes not encountered during the training phase. Methods for novelty detection can be categorized into two main groups of open-set recognition (OSR)

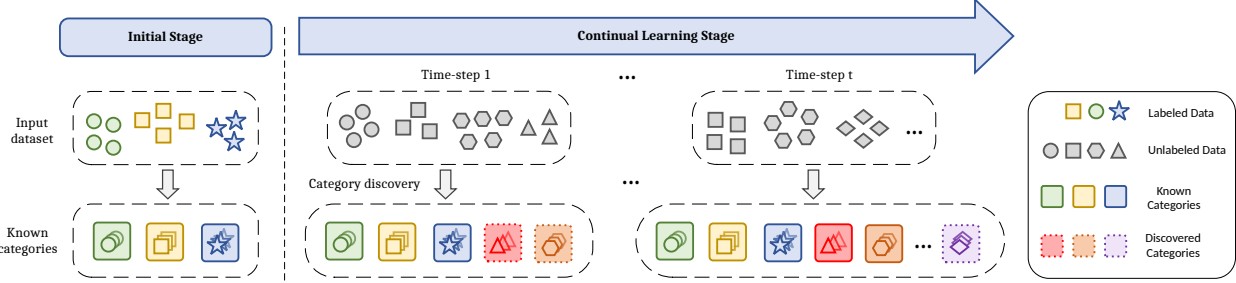

Figure 1: The general presentation of the Continual Generalized Category Discovery (CGCD) setting. In the initial stage, a labeled dataset is provided to train the initial model. After the initial stage, the model enters the continual learning stage, in which no labeled data is provided. The input data in this stage can contain samples belonging to novel or previously known categories. The model is expected to discover potential novel categories in this unlabeled data and integrate them into the model without compromising the performance of previous categories and making assumptions about the number of novel categories.

and out-of-distribution (OOD) detection. Despite having small differences, these methods share the goal of detecting samples from unknown categories. OOD detection methods include post-hoc, reconstruction, distance-based, density-based, and training-staged approaches. Post-hoc methods do not change the training process Hendrycks & Gimpel (2016); Liang et al. (2017); Hendrycks et al. (2019); Liu et al. (2020); Sun et al. (2021); Sun & Li (2022). Reconstruction methods leverage the fact that models trained on ID data cannot effectively compress or reconstruct OOD data Denouden et al. (2018); Jiang et al. (2023). Distance-based methods use distance measures to ID centroids or prototypes Lee et al. (2018); Sun et al. (2022); Wang et al. (2022); Ming et al. (2022); Kim et al. (2024). Density-based methods rely on probabilistic modeling to flag samples in low-density regions as OOD Abati et al. (2019); Jiang et al. (2021); Zisselman & Tamar (2020). Training-staged methods modify the training process to separate ID and OOD data Hendrycks et al. (2018); Yu & Aizawa (2019); Lee et al. (2017a); Du et al. (2022); Hsu et al. (2020).

Open-set recognition methods are categorized as discriminative, generative, or hybrid. Discriminative classifiers are extended to perform open-set recognition either using a distance measure Shu et al. (2020); Hassen & Chan (2020); Cevikalp (2016) or constructing probability functions for ID/OOD distinction Scherreik & Rigling (2016); Shu et al. (2017); Rudd et al. (2017); Yoshihashi et al. (2019); Oza & Patel (2019). Generative models create synthetic unknown samples for training Ge et al. (2017); Neal et al. (2018). Hybrid approaches combine discriminative and generative strategies to learn embeddings jointly Zhang et al. (2020); Zhou et al. (2021); Chen et al. (2021).

**Category discovery** methods try to identify novel categories within unlabeled data. Early approaches assume that unlabeled data contains only novel samples and typically follow a multi-stage process, involving pre-training and fine-tuning on labeled and unlabeled data. These approaches include generating pair-wise pseudo-labels based on feature magnitudes Han et al. (2021); Zhao & Han (2021); Zhong et al. (2021), clustering unlabeled data using deep embedding clustering Xie et al. (2016); Han et al. (2019), and assigning pseudo-labels through self-labeling Joseph et al. (2022).

Recent advancements have addressed more realistic scenarios through Generalized Category Discovery (GCD), where unlabeled data includes both previously known and novel classes. GCD Vaze et al. (2022) combines self-supervised pre-training, contrastive learning-based fine-tuning, and semi-supervised k-means clustering for novel class discovery. Gaussian mixture model for generalized category discovery with Protypical Contrastive learning (GPC) Zhao et al. (2023a) employs a semi-supervised Gaussian Mixture Model with prototypical contrastive learning, enabling joint representation learning, cluster estimation, and label assignment without assuming a fixed number of clusters. In contrast to the multi-stage methods, UNified Objective function (UNO) Fini et al. (2021) presents a unified objective function that simultaneously addresses feature extraction, labeled data classification, and clustering of unlabeled data by leveraging pseudo-labels and employing multiple clustering and over-clustering strategies. Cross-instance Positive Relations (CiPR) Hao

et al. (2024) proposes a semi-supervised hierarchical clustering algorithm to obtain reliable cross-instance relations in order to bootstrap the representation in contrastive learning.

**Continual learning** methods aim to address the challenge of catastrophic forgetting, enabling models to retain previously learned knowledge, while adapting to new tasks. Existing approaches can be broadly categorized into the following: Replay-based techniques focus on mitigating forgetting by storing a subset of previous task data and replaying it during training to reinforce memory of the earlier tasks Chaudhry et al. (2019); Rolnick et al. (2019); Zheng et al. (2024); Lin et al. (2023). Regularization-based techniques that aim to preserve prior knowledge by guiding the training process, including parameter regularization to prevent critical parameters from significant updates Kirkpatrick et al. (2017); Lee et al. (2017b); Zenke et al. (2017); Wistuba et al. (2023), knowledge distillation, where a "teacher" model transfers knowledge to a "student" model Li & Hoiem (2017); Wu et al. (2018); Hou et al. (2019), and data regularization that constrains the optimization process using stored exemplars Lopez-Paz & Ranzato (2017); Chaudhry et al. (2018); Zhao et al. (2023b). Meta-learning approaches aim to train models capable of rapidly adapting to new tasks with minimal data, leveraging learned algorithms for efficient generalization Finn et al. (2017); Gupta et al. (2018); Ostapenko et al. (2019); Saha & Roy (2025). Architecture-based methods address continual learning by modifying or expanding the model's structure to accommodate new tasks, while preserving existing knowledge Yoon et al. (2017); Rusu et al. (2016); Hu et al. (2023). Hybrid methods that use the combination of the aforementioned techniques Li et al. (2019); Wu et al. (2021); Douillard et al. (2020).

**Continual generalized category discovery** methods aim to identify novel categories in unlabeled data, which may include both novel samples and samples from previously known classes. These methods tackle the challenge of discovering new categories incrementally, while retaining knowledge of existing ones. Grow and Merge (GM) Zhang et al. (2022) introduces a framework with two models: a static model that preserves knowledge of old classes and a dynamic model trained in a self-supervised manner to discover novel classes. Novel samples are identified based on their distance to the prototypes of known classes, and the two models are merged, when a new task arrives. Incremental Generalized Category Discovery (IGCD) Zhao & Mac Aodha (2023) employs a non-parametric classifier combined with a density-based exemplar selection method that is employed to select exemplars samples as well discovering novel classes. Proxy Anchor (PA) Kim et al. (2023) uses proxies learned through proxy anchor loss Kim et al. (2020) to detect novel samples. A non-parametric clustering algorithm clusters and identifies new categories, while catastrophic forgetting is mitigated through experience replay and knowledge distillation. MetaGCD Wu et al. (2023) adopts a meta-learning framework that leverages offline labeled data, to simulate the incremental learning process and utilizes the classic k-means algorithm for novel class discovery. Happy Ma et al. (2024) identifies prediction bias and hardness bias as the main challenges in the CGCD scenario, and proposes to address them through clustering-guided initialization and soft entropy regularization for mitigating prediction bias, along with hardness-aware prototype sampling to tackle hardness bias.

## 3 Problem Definition

This section presents the problem setting. In the framework of Continual Generalized Category Discovery (CGCD), the model begins with an initial labeled dataset $D_l^0 = \{(x, y) \in \mathcal{X}_l^0 \times \mathcal{Y}_l^0\}$, where $\mathcal{X}_l^0$ represents the labeled input data and $\mathcal{Y}_l^0$ denotes the corresponding labels. This dataset is used to train a feature extractor $\mathcal{F}^0 : \mathcal{X} \to \mathcal{Z}^0$, which generates embedding vectors $\mathcal{Z}^0$, and a classifier $\mathcal{C}^0 : \mathcal{Z}^0 \to \mathcal{Y}^0$, which maps the embeddings to specific classes.

Following this initial phase, no additional labeled data is provided. Instead, the model is sequentially exposed to a series of unlabeled datasets, $\{D_u^t\}_{t=1}^T$, where $D_u^t = \{x \in \mathcal{X}_u^t\}$. Here, $\mathcal{X}_u^t$ contains the unlabeled input data and $T$ denotes the number of time steps. These datasets include examples from previously encountered categories as well as samples from potentially new categories, i.e., $\mathcal{Y}_l \cap \mathcal{Y}_u \neq \emptyset$. No assumption is made about the presence or the number of novel categories in the unlabeled data.

The model's task is to identify and learn new categories, while updating the feature extractor $\mathcal{F}^t : \mathcal{X} \to \mathcal{Z}^t$ and classifier $\mathcal{C}^t : \mathcal{Z}^t \to \mathcal{Y}^t$ at each time step $t$. Over time, $\mathcal{Y}^t$ encompasses all categories seen so far, i.e., $\mathcal{Y}^t = \mathcal{Y}^{t-1} \cup \mathcal{Y}_{new}^t$, where $\mathcal{Y}_{new}^t$ is the newly discovered categories in the current time step $t$. Figure 1 illustrates this setting.

The key challenge lies in balancing the catastrophic forgetting of previously learned data (stability) with the effective discovering and learning of new categories (plasticity).

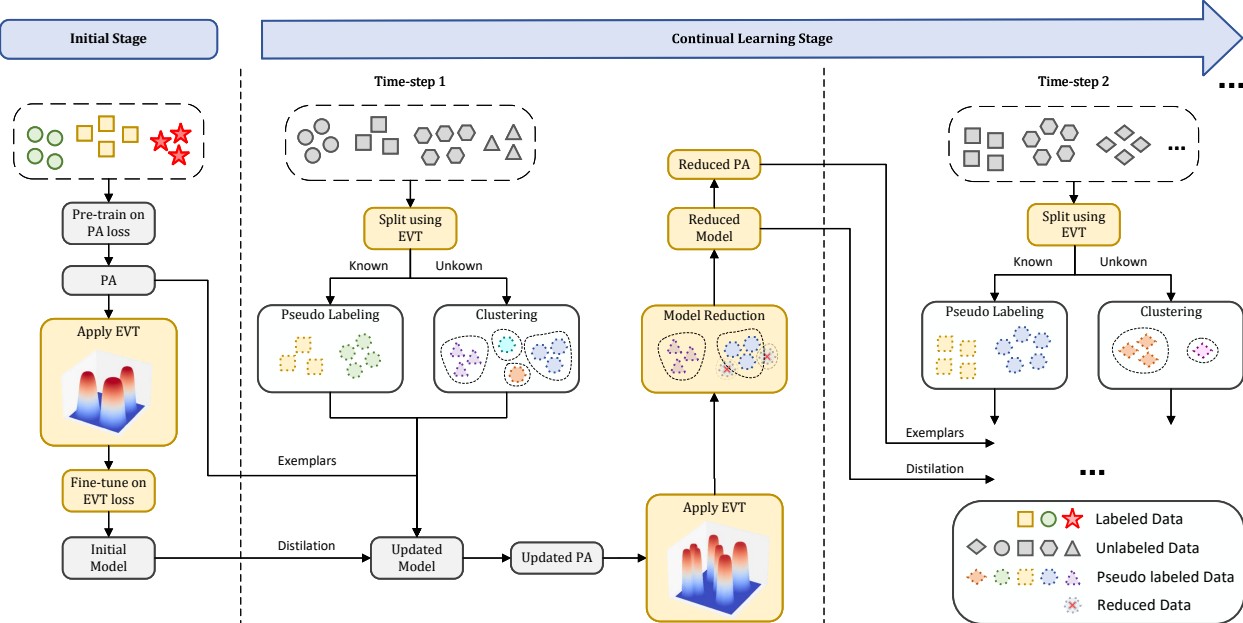

Figure 2: Overview of CATEGORIZER. In the initial stage, the model is first pre-trained on PA loss to derive proxy anchors for different classes. Following this, the EVT analysis is applied to each proxy to compute the Weibull distribution around each proxy and devising a probability of inclusion (PSI) function that is capable of rejecting unknown samples. With the computed distributions, we fine-tune the model on our novel *evt* loss to get the initial model. In the continual learning stage, the input data containing novel and known samples are separated by thresholding PSIs functions computed in the initial stage into known and unknown samples. Known samples are pseudo-labeled using the current model from the previous step and unknown samples are clustered. The model is updated using pseudo-labeled and clustered data, exemplars of previous categories, and distillation loss derived from the previous step model. EVT is applied to the updated model to get updated distribution, where the model is reduced and redundant proxies are discarded. This process repeats for the next steps. The yellow boxes indicate our novel contribution in the proposed scheme.

## 4 Proposed Method

Our proposed novel method is composed of two main stages: 1) the initial stage and 2) the continual learning stage. In the initial stage, the feature extractor $\mathcal{F}^0$ and classifier $\mathcal{C}^0$ are trained using the labeled dataset. In the subsequent continual learning stage, the process begins with a novelty detection module that divides the data into known and unknown sets. The unknown samples are then subjected to a novel class discovery procedure, which identifies coherent and distinct samples as potential new classes. Finally, the feature extractor and classifier are updated to incorporate the newly discovered samples. Figure 2 illustrates the overview of CATEGORIZER. The following sections provide detailed explanations of the main steps involved.

### 4.1 Initial Stage

To train the feature extractor $\mathcal{F}^0$, we use the Proxy Anchor (PA) loss Kim et al. (2020) to pre-train the model. This PA loss is a deep metric learning loss, which is known for its strong performance and fast convergence. Moreover, introducing new classes is as straightforward as adding new proxies. Specifically, it

leverages data-to-data and data-to-proxy relationships to minimize the following loss function:

$$\ell_{pa}(Z) = \frac{1}{|P^+|} \sum_{p \in P^+} \log \left( 1 + \sum_{z \in Z_p^+} e^{-\alpha(s(z,p)-\delta)} \right)$$
$$+ \frac{1}{|P^-|} \sum_{p \in P^-} \log \left( 1 + \sum_{z \in Z_p^-} e^{\alpha(s(z,p)+\delta)} \right), \tag{1}$$

where $\delta > 0$ is a margin, $\alpha > 0$ is a scaling factor, $s(\cdot, \cdot)$ is the similarity function, $P$ represents the set of all proxies, and $P^+$ and $P^-$ are the set of positive and negative proxies in the batch, respectively. For each proxy $p$, the batch of embedding vectors $Z$ is split into two sets: $Z_p^+$ stands for positive embedding vectors of $p$ and $Z_p^- = Z \setminus Z_p^+$.

---

**Algorithm 1** Initial Stage Training Session.

---

**Input:** Training dataset $D^0$, learning rate $\eta$, number of epochs $E$, batch size $B$, tail size $\tau$, and threshold $\epsilon$.
**Output:** Feature extractor $\mathcal{F}^0$ and classifier $\mathcal{C}^0$.
**Initialize:** Proxy set $P$, feature extractor $\mathcal{F}$ with random weights.
**Step 1: Pre-training on PA loss**
**for** epoch $e = 1$ to $E$ **do**
    **for** each mini-batch $\mathcal{B} \subset D^0$ of size $B$ **do**
        Compute PA loss $\ell_{pa}(\mathcal{B})$ (Eq.1).
        Update feature extractor $\mathcal{F}$ as:
$$\mathcal{F} \leftarrow \mathcal{F} - \eta \frac{\partial \ell_{pa}(\mathcal{B})}{\partial \mathcal{F}}.$$

    **end for**
**end for**
**Step 2: Apply EVT**
**for** each proxy $p$ **do**
    Estimate Weibull parameters $\lambda_p$ and $\kappa_p$ using the tail size $\tau$.
    Construct probability of inclusion function $\Psi$ (Eq.2).
**end for**
Construct classifier $\mathcal{C}^0$ using calculated inclusion functions and threshold $\epsilon$ (Eq.5)
**Step 3: Fine-tuning on** *evt* **loss**
**for** epoch $e = 1$ to $E$ **do**
    **for** each mini-batch $\mathcal{B} \subset D^0$ of size $B$ **do**
        Compute *evt* loss $\ell_{evt}(\mathcal{B})$ (Eq.3).
        Update feature extractor $\mathcal{F}$ as:
$$\mathcal{F} \leftarrow \mathcal{F} - \eta \frac{\partial \ell_{evt}(\mathcal{B})}{\partial \mathcal{F}}.$$

    **end for**
**end for**
**Return** $\mathcal{F}^0$ and $\mathcal{C}^0$.

---

After pretraining, we apply Extreme Value Theory (EVT) to enhance the learned representations and improve the model's ability to detect novel samples. For each proxy $p_i \in P$, we identify the $\tau$ nearest samples, $N_{p_i} = \{(x_j, y_j)\}_{j=1}^\tau$, where $y_{p_i} \neq y_j$, and calculate their distances in the embedding space $m_{ij} = s(z_j, p_i)$ to the proxy $p_i$. The distribution of these minimal distances follows a Weibull distribution based on EVT Rudd et al. (2017). The probability of a sample $x'$ being within the boundary of proxy $p_i$ is given by the inverse Weibull distribution:

$$\Psi(p_i, z'; \kappa_i, \lambda_i) = e^{-\left( \frac{s(p_i, z')}{\lambda_i} \right)^{\kappa_i}}, \tag{2}$$

where $s(p_i, z')$ is the similarity between $z'$ and proxy $p_i$, $z'$ is the embedding vector of sample $x'$ and $\kappa_i, \lambda_i$ are the Weibull shape and scale parameters estimated from the smallest $m_{ij}$ values Rudd et al. (2017). This function provides the probability of sample inclusion (PSI) that is capable of rejecting unknown samples.

So far, EVT has been applied as a post-hoc procedure, meaning it does not affect the learned representation, potentially leading to suboptimal results. To address this, we propose a novel *evt* loss to fine-tune the model based on the estimated Weibull distribution:

$$\ell_{evt}(Z) = \frac{1}{|P^+|} \sum_{p \in P^+} \log \left( 1 + \sum_{z \in Z_p^+} \left( 1 - e^{-\left( \frac{s(z,p)}{\lambda_p} \right)^{\kappa_p}} \right) \right)$$
$$+ \frac{1}{|P^-|} \sum_{p \in P^-} \log \left( 1 + \sum_{z \in Z_p^-} e^{-\left( \frac{s(z,p)}{\lambda_p} \right)^{\kappa_p}} \right), \tag{3}$$

where $\kappa_p, \lambda_p$ are the Weibull shape and scale parameters for proxy $p$. The first term encourages higher probabilities for positive proxies with respect to positive samples, while the second term penalizes high probabilities for negative proxies. This fine-tuning, in addition to improving the capability of the model for unknown detection, enhances the learned representation compared to the plain PA loss (see Table 3).

Using Eq. (2), the probability of an input $x'$ belonging to class $l$ is computed as:

$$\hat{P}(l \mid z') = \max_{\{i:y_i=l\}} \Psi(p_i, z'; \kappa_i, \lambda_i) \tag{4}$$

Using this, the classifier $\mathcal{C}^0$ is defined as:

$$\mathcal{C}^0 = \begin{cases} \arg\max_{l \in \{1,\dots,M^0\}} \hat{P}(l \mid z'), & \text{if } \hat{P}(l \mid z') \geq \epsilon, \\ \text{``unknown''}, & \text{otherwise}, \end{cases} \tag{5}$$

where $M^0$ is the total number of classes in the initial stage, and $\epsilon$ is a threshold for rejecting unknown samples. A practical way to select $\epsilon$ is to optimize the trade-off between closed-set accuracy and the rejection of unknown classes through cross-class validation. In our experiments, a common threshold worked well enough across different datasets. See Algorithm 1 for the pseudo-code for training the model in the initial stage.

## 4.2 Continual Learning Stage

After the initial stage and training on the labeled data, the model enters the continual learning stage, where no label will be provided and the input data might contain samples from both novel categories and known categories. The data first goes through a novelty detection module, where it is split into known and unknown samples. The detected unknown samples are then examined using the novel class discovery algorithm to find novel categories. In the end, the model is updated based on these new categories.

### 4.2.1 Novelty Detection

We utilize the classifier, which was trained using EVT in the initial stage to perform novelty detection. More specifically at time $t$, we use the classifier learned at the previous step $\mathcal{C}^{t-1}$ to separate unknown samples, as well as assign pseudo labels to known samples

$$\mathcal{C}^{t-1} = \begin{cases} \arg\max_{l \in \{1,\dots,M^{t-1}\}} \hat{P}(l \mid z'), & \text{if } \hat{P}(l \mid z') \geq \epsilon, \\ \text{``unknown''}, & \text{otherwise}, \end{cases} \tag{6}$$

where $M^{t-1}$ is the total number of classes accumulated until time $t-1$, and $\epsilon$ is the threshold for rejecting unknown samples as discussed before. This way the data is split into two sets $D_{\text{known}}^t$ and $D_{\text{unknown}}^t$, where $D_{\text{known}}^t$ is the set of samples belonging to previously known categories (i.e., pseudo labeling) and $D_{\text{unknown}}^t$ are samples of novel categories (i.e., input for novelty detection), where $D^t = D_{\text{known}}^t \cup D_{\text{unknown}}^t$

### 4.2.2 Novel Class Discovery

The separated unknown samples $D_{\text{unknown}}^t$ contain potential novel categories that need to be further discovered. To do this the most common approach is clustering these unknown data. Since many clustering algorithms need to know the number of clusters, and estimating this number can add more complexity to the algorithm, we follow Kim et al. (2023) by using a non-parametric clustering algorithm. Specifically, we use the affinity propagation method Frey & Dueck (2007) to cluster and discover novel classes. In our experiments, we observed that using affinity propagation for clustering in our proposed scheme leads to over-clustering, which leads to overestimating the number of novel categories, degrading the model's performance in both novel categories and known categories. This problem also exists in other common clustering algorithms (see Appendix A), and is mitigated in the following class incremental learning step.

---

**Algorithm 2** Continual Learning Stage.

---

1: **Input:** Unlabeled data $D_u^t$, classifier $\mathcal{C}^{t-1}$, feature extractor $\mathcal{F}^{t-1}$, proxy set $P^{t-1}$, tail size $\tau$, learning rate $\eta$, threshold $\epsilon$, number of epochs $E$, and batch size $B$.
2: **Output:** Updated feature extractor $\mathcal{F}^t$, updated classifier $\mathcal{C}^t$, and updated proxy set $P^t$.
3: **Step 1: Split Unlabeled Data**
4: Use classifier $\mathcal{C}^{t-1}$ (Eq.6) to split $D_u^t$ into:

$$D_{\text{known}}^t \quad \text{and} \quad D_{\text{unknown}}^t.$$

5: **Step 2: Process Known and Unknown Data**
6: **Pseudo-labeling:** For $x \in D_{\text{known}}^t$, assign pseudo-labels using $\mathcal{C}^{t-1}$.
7: **Clustering:** Apply clustering on $D_{\text{unknown}}^t$ to group data points into clusters $C_i$.
8: Initialize the proxy set $P^t \leftarrow P^{t-1}$
9: **for** each cluster $C_i$ in $D_{\text{unknown}}^t$ **do**
10:     Initialize a new proxy $p_i$ at centroid of $C_i$ and add it to the proxy set $P^t$:

$$P^t \leftarrow \{P^t \cup p_i\}.$$

11: **end for**
12: **Step 3: Train Feature Extractor**
13: **for** epoch $e = 1$ to $E$ **do**
14:     **for** each mini-batch $\mathcal{B} \subset D_u^t$ of size $B$ **do**
15:        Compute the total loss $\mathcal{L}^t$ as:

$$\mathcal{L}^t = \mathcal{L}_{pa}^t(Z^t) + \mathcal{L}_{fr}^t(\tilde{Z}) + \mathcal{L}_{kd}^t(z_o)$$

    (See Eqs.1, 8, and 7).
16:        Update feature extractor $\mathcal{F}^t$ as:

$$\mathcal{F}^t \leftarrow \mathcal{F}^{t-1} - \eta \frac{\partial \mathcal{L}^t}{\partial \mathcal{F}}.$$

17:     **end for**
18: **end for**
19: **Step 4: Recalculate Weibull Distributions and Construct Inclusion Functions**
20: **for** each proxy $p \in P^t$ **do**
21:     Estimate Weibull parameters $\lambda_p$ and $\kappa_p$ using tail size $\tau$.
22: **end for**
23: **Step 5: Model Reduction**
24: Remove redundant proxies of newly discovered categories from the proxy set $P^t$ based on the updated Weibull distributions (Algorithm 3).
25: **Return** Updated feature extractor $\mathcal{F}^t$, updated classifier $\mathcal{C}^t$, and updated proxy set $P^t$.

---

### 4.2.3 Class Incremental Learning

The known samples in the data have been assigned pseudo labels by the previous classes and novel samples have their clustering number as pseudo class. Given this, we can integrate the newly discovered categories into the model. To do this we create new proxies for each discovered cluster, i.e., new category and initialize the proxies on the centroids of clusters, following Kim et al. (2023). To achieve this we add the set of proxies of new classes $P_{new}$ to set of proxies $P^t = \{P^{t-1} \cup P_{\text{new}}\}$ and optimize the same proxy anchor loss as in Eq. (1) to improve the performance of the model on the discovered novel categories.

To avoid catastrophic forgetting during this update, the well-known problem of continual learning, we use feature distillation and feature replay methods. More specifically the distillation loss for feature distillation is defined as

$$\mathcal{L}_{kd}^t(z_o) = -\mathbb{E}_{x_o \in D_{\text{known}}^t} \left\| \mathcal{F}^{t-1}(x_o) - \mathcal{F}^t(x_o) \right\|_2, \tag{7}$$

Furthermore, to improve robustness against catastrophic forgetting, we sample cosine distances from each proxy's Weibull distribution and use spherical interpolation to place unit-norm exemplars on the hypersphere. This preserves the intrinsic geometry of cosine space and aligns more closely with the feature space learned by the EVT loss, compared to alternative distributions such as the Gaussian used in Kim et al. (2023) (see Table 8 for a comparison with Gaussian-based sampling). Based on these synthesized features, we define the feature replay loss as follows:

$$\mathcal{L}fr^t(\tilde{Z}) = \mathcal{L}pa^t(\tilde{Z}), \quad \tilde{Z} = \{\tilde{z} \mid \tilde{z} = \text{SphInterp}(p^{t-1}, d), \ d \sim \text{Weibull}(\kappa^{t-1}, \lambda^{t-1})\} \tag{8}$$

The overall loss for updating the model can be defined as

$$\mathcal{L}^t = \mathcal{L}_{pa}^t(Z^t) + \mathcal{L}_{fr}^t(\tilde{Z}) + \mathcal{L}_{kd}^t(z_o) \tag{9}$$

where one loss optimizes the model based on the current data for discovered categories and two other losses prevent catastrophic forgetting.

After training the model on the Eq. (9), i.e., getting updated feature extractor $\mathcal{F}^t$, we derive the classifier $\mathcal{C}^t$ using the same approach that was used in the initial stage, i.e., using EVT and fitting the Weibull distribution using $\tau$ nearest points of opposite class samples for each proxy. The overall pseudo code of the continual learning stage is provided in Algorithm 2

As was mentioned in section 4.2.2, the clustering of novel samples leads to an over-clustering of the data, i.e., a category might have more than one cluster. After training the model on Eq. (9), the clusters of the same category are likely to be pushed near each other, we employ this fact to reduce the number of proxies following the approach in Rudd et al. (2017) to remove redundant proxies. Let $p_i$ be a proxy of a discovered novel class and $\Psi(p_i, p', \kappa_i, \lambda_i)$ be its corresponding fitted Weibull distribution. To decide the redundancy of the pair $\langle p_i, \Psi(p_i, p', \kappa_i, \lambda_i) \rangle$, i.e., whether to keep it, we define an indicator function $I(\cdot)$ such that

$$I(p_i) = \begin{cases} 1 & \text{if } \langle p_i, \Psi(p_i, p', \kappa_i, \lambda_i) \rangle \text{ is kept,} \\ 0 & \text{otherwise.} \end{cases} \tag{10}$$

Then, we can define the optimization problem of selecting a minimum number of proxies such that each proxy is at least covered by another proxy as follows:

$$\begin{cases} \text{minimize } \sum_{i=1}^N I(p_i) \\ \text{subject to } \forall j \exists i \mid I(p_i)\Psi(p_i, p_j, \kappa_i, \lambda_i) \geq \zeta. \end{cases} \tag{11}$$

where $N$ is the number of discovered classes and $\zeta$ is the threshold used to determine whether a proxy is covered by another proxy. In our experiments, we set this threshold to a very high value near 1 to only

---

**Algorithm 3** Model Reduction using Greedy Set Cover.

---

**Input:** Proxy set $P = \{p_1, p_2, \ldots, p_N\}$, corresponding Weibull parameters $(\lambda_i, \kappa_i)$, inclusion threshold $\zeta$.
**Output:** Reduced proxy set $P_{\text{reduced}}$.
**Step 1: Compute Subsets**
For each proxy $p_i \in P$, compute the set of covered proxies (Eq.11):

$$S_i = \{p_j \mid I(p_i)\Psi(p_i, p_j, \kappa_i, \lambda_i) \geq \zeta\}$$

**Step 2: Initialize Greedy Set Cover**
Define the universe as all proxies $U = P$, and subsets as $\{S_1, S_2, \ldots, S_N\}$.
Initialize:
covered $\leftarrow \emptyset$
$P_{\text{reduced}} \leftarrow \emptyset$
**Step 3: Greedy Selection**
**while** covered $\neq U$ **do**
    Find the subset $S_i$ that covers the maximum uncovered proxies:

$$i^* = \arg\max_i |S_i \setminus \text{covered}|.$$

    Add proxy $p_{i^*}$ to the reduced proxy set:

$$P_{\text{reduced}} \leftarrow \{P_{\text{reduced}} \cup p_{i^*}\}.$$

    Update the covered set:

$$\text{covered} \leftarrow \{\text{covered} \cup S_{i^*}\}.$$

**end while**
**Return** Reduced proxy set $P_{\text{reduced}}$.

---

reduce proxies that are very close to each other. Since the optimization problem in Eq. (11) is a special case of the Karp's Set Cover problem Rudd et al. (2017) and is NP-hard, we follow the greedy approach introduced in Slavík (1996) to solve this problem approximately. We begin with defining the universe $U$ as all the proxies, initializing the covered set as an empty set, and finding subsets of covered proxies of each proxy based on Eq.10. In each iteration, we select the subset that covers the most uncovered proxies, add this subset to the covered set, and repeat this process until all proxies are covered (See Algorithm 3).

## 5 Expermiental Results

This section provides implementation details and evaluation metrics, and, then, compares the results obtained by the proposed novel method with those obtained through state-of-the-art works. It finally presents the ablation study for the proposed method.

### 5.1 Implementation Details

For the sake of a fair comparison, we utilized ResNet18 He et al. (2016), pre-trained on ImageNet-1k, as the feature extractor across all methods. For the data augmentation, we employed commonly used techniques such as random crops and horizontal flips. For the proxy anchor loss hyperparameters, $\alpha$ is set to 0.1, and $\delta$ is set to 32. The hyperparameters $\tau$, $\epsilon$, and $\zeta$ of CATEGORIZER are set to 500, 0.75, and 0.999, respectively. The hyperparameter analysis of $\tau$ and $\epsilon$ is shown in Table 1 and Table 2, respectively. This analysis shows that the sensitivity with respect to these two hyperparameters is low, and a common value has worked well across different datasets. Since $\zeta$ is only used during the continual stage and can't be configured in the initial stage, we set it to a high value to make it robust across different datasets.

Table 1: Initial accuracy of the model using different number of neighbours to be used in the EVT modeling. The results show low sensitivity towards this hyperparameter.

| DATASET | 100 | 250 | 500 | 1000 | 2000 |
|---------|-----|-----|-----|------|------|
| CUB | 80.72 | 80.91 | **80.98** | 80.96 | 80.76 |
| MIT | 72.54 | 72.45 | **72.83** | 72.26 | 72.54 |
| DOGS | 84.86 | 84.70 | **84.77** | 84.67 | 84.76 |
| CIFAR10 | 96.07 | 96.08 | **96.07** | 96.15 | 96.12 |
| CIFAR100 | 80.90 | 80.86 | **80.98** | 80.83 | 80.95 |
| AIRCRAFT | 65.19 | 65.22 | **65.49** | 65.16 | 65.01 |
| SVHN | 96.50 | 96.53 | 96.53 | **96.54** | 96.51 |
| MNIST | 99.00 | 98.99 | **99.03** | 99.02 | 98.99 |
| FASHION-MNIST | 97.60 | **97.58** | 97.55 | 97.56 | 97.56 |
| CARS | 67.85 | 67.86 | **68.30** | 68.21 | 68.13 |

Table 2: Accuracy of the novelty detection module using different thresholds. The result indicates low sensitivity towards this hyperparameter

| DATASET | 0.3 | 0.5 | 0.75 | 0.95 |
|---------|-----|-----|------|------|
| CUB | 63.76 | 69.69 | **70.57** | 66.03 |
| MIT | 63.82 | 66.19 | **67.20** | 63.67 |
| DOGS | 61.64 | 65.55 | **68.74** | 65.05 |
| CIFAR10 | 62.50 | 66.00 | **70.00** | 64.50 |
| CIFAR100 | 62.30 | 66.10 | **69.10** | 64.80 |
| AIRCRAFT | 61.90 | 65.70 | **70.20** | 64.60 |
| SVHN | 62.00 | 66.30 | **69.50** | 64.70 |
| MNIST | 62.20 | 66.40 | **68.30** | 64.40 |
| FASHION-MNIST | 61.80 | 65.80 | **70.10** | 64.30 |
| CARS | 70.19 | 72.24 | **75.32** | 69.83 |

We used an RTX 3070 GPU, Ryzon 5 7600x CPU, and 32GB RAM for running the experiments. During the initial training stage, the model was trained using the AdamW optimizer with a weight decay of 0.0001 and a learning rate of 0.0001 for 60 epochs on PA loss Kim et al. (2020) and 60 epochs on our proposed *evt* loss. The learning rate was reduced by half every five epochs. In the continual learning stage, the model was updated over 10 epochs. In our experiments, we observed that training beyond this point significantly degrade the performance on novel categories in some datasets (See Figure 5). To fit the Weibull distributions, we used a torch implementation based on Vastlab (2024) to estimate the shape and scale parameters of a Weibull distribution. The overhead of this process is negligible, making it fast to compute. For other methods, we adhered to the hyperparameters and network architectures specified in their original implementations, referring to the respective papers for details. All reported results represent the average performance over all runs.

## 5.2 Evaluation Metrics

We employ a clustering accuracy-based measurement, called Hungarian assignment algorithm Kuhn (1955), to measure the performance of the model, following previous methods Zhang et al. (2022); Kim et al. (2023); Zhao & Mac Aodha (2023); Wu et al. (2023). This metric is defined as

$$\mathcal{M}^t = \frac{1}{|\mathcal{D}|} \sum_{i=1}^{|\mathcal{D}|} \mathbb{I}(y_i = h^*(y_i^*)), \tag{12}$$

where $|\mathcal{D}|$ is the size of the evaluation dataset, $\mathbb{I}$ is the indicator function, $y_i$ and $y_i^*$ are the ground-truth label and clustering assignment for the i-th sample, and $h^*$ is the optimal assignment. This algorithm aligns predicted clusters with true labels to accurately evaluate the clustering performance. Based on this, we use $M_{all}$ and $M_o$ to measure the clustering accuracy on all of the categories and old categories, respectively. To measure the performance drop in the previously known classes after discovering and learning new categories, i.e., forgetting, we employ $M_f$ metric, which is proposed in Zhang et al. (2022) and defined as

$$\mathcal{M}_f = \max_t \{\mathcal{M}_o^0 - \mathcal{M}_o^t\}, \tag{13}$$

where $\mathcal{M}_o^0$ and $\mathcal{M}_o^t$ are the clustering accuracy of old categories, i.e., known categories, at the initial stage and time step $t$.

Table 3: Performance comparison w.r.t. the Recall@K metric on the model trained with different methods on fine-grained datasets. GM Zhang et al. (2022) uses traditional cross-entropy loss, PA Kim et al. (2023) uses proxy anchor loss Kim et al. (2020), and IGCD Zhao & Mac Aodha (2023), MetaGCD Wu et al. (2023) combine a supervised contrastive loss from Khosla et al. (2020) and unsupervised contrastive loss from Chen et al. (2020) and Gutmann & Hyvärinen (2010), respectively, and Happy Ma et al. (2024) combines cross-entropy loss, supervised Khosla et al. (2020) and unsupervised contrastive loss Chen et al. (2020). Our proposed evt loss outperforms all mentioned SOTA methods in almost all datasets.

| Dataset | Method | R@1 | R@2 | R@4 | R@8 |
|---|---|---|---|---|---|
| CUB | GM | 55.83 | 67.36 | 75.80 | 83.20 |
| | MetaGCD | 59.61 | 69.91 | 78.97 | 83.20 |
| | PA | 70.57 | 79.47 | 86.50 | 91.16 |
| | IGCD | 64.01 | 73.85 | 81.74 | 87.36 |
| | Happy | 42.23 | 55.28 | 67.22 | 77.35 |
| | **CATEGORIZER** | **73.69** | **81.17** | 87.22 | **91.66** |
| MIT | GM | 54.70 | 65.37 | 73.06 | 82.23 |
| | MetaGCD | 59.32 | 70.82 | 78.50 | 84.47 |
| | PA | 58.80 | 69.70 | 79.70 | 87.01 |
| | IGCD | 63.06 | 72.01 | 79.10 | 84.77 |
| | Happy | 31.79 | 43.20 | 54.92 | 66.94 |
| | **CATEGORIZER** | **65.47** | **73.77** | **80.56** | **88.01** |
| Dogs | GM | 48.20 | 61.71 | 72.64 | 80.49 |
| | MetaGCD | 62.65 | 74.15 | 83.72 | 89.16 |
| | PA | 75.77 | 84.06 | **91.42** | **95.60** |
| | IGCD | 64.45 | 74.46 | 82.02 | 87.48 |
| | Happy | 61.24 | 75.36 | 85.30 | 91.88 |
| | **CATEGORIZER** | **77.55** | **84.96** | 90.23 | 93.49 |
| Cifar100 | GM | 51.29 | 61.74 | 71.75 | 79.74 |
| | MetaGCD | 52.74 | 63.58 | 72.86 | 80.96 |
| | PA | 76.13 | 82.48 | 88.06 | 92.21 |
| | IGCD | 77.70 | 83.02 | 86.71 | 89.30 |
| | **CATEGORIZER** | **79.06** | **83.92** | 87.18 | 89.90 |
| Cars | GM | 72.60 | 81.27 | 87.50 | 91.64 |
| | MetaGCD | 59.44 | 71.57 | 81.33 | 88.39 |
| | PA | 69.66 | 79.12 | 86.71 | 92.42 |
| | IGCD | 77.08 | 85.13 | 90.41 | 94.03 |
| | Happy | 60.64 | 73.81 | 83.09 | 90.23 |
| | **CATEGORIZER** | **78.02** | **85.64** | **91.08** | **94.49** |

To measure the ability of the model to discover and learn novel categories, we employ $M_d$ measure Zhang et al. (2022), which is described as

$$\mathcal{M}_d = \frac{1}{|T|} \sum_{i=T} \mathcal{M}_n^i. \tag{14}$$

where $M_n^i$ is the clustering accuracy on novel categories at the $i$-th step and $T$ is the total number of learning steps.

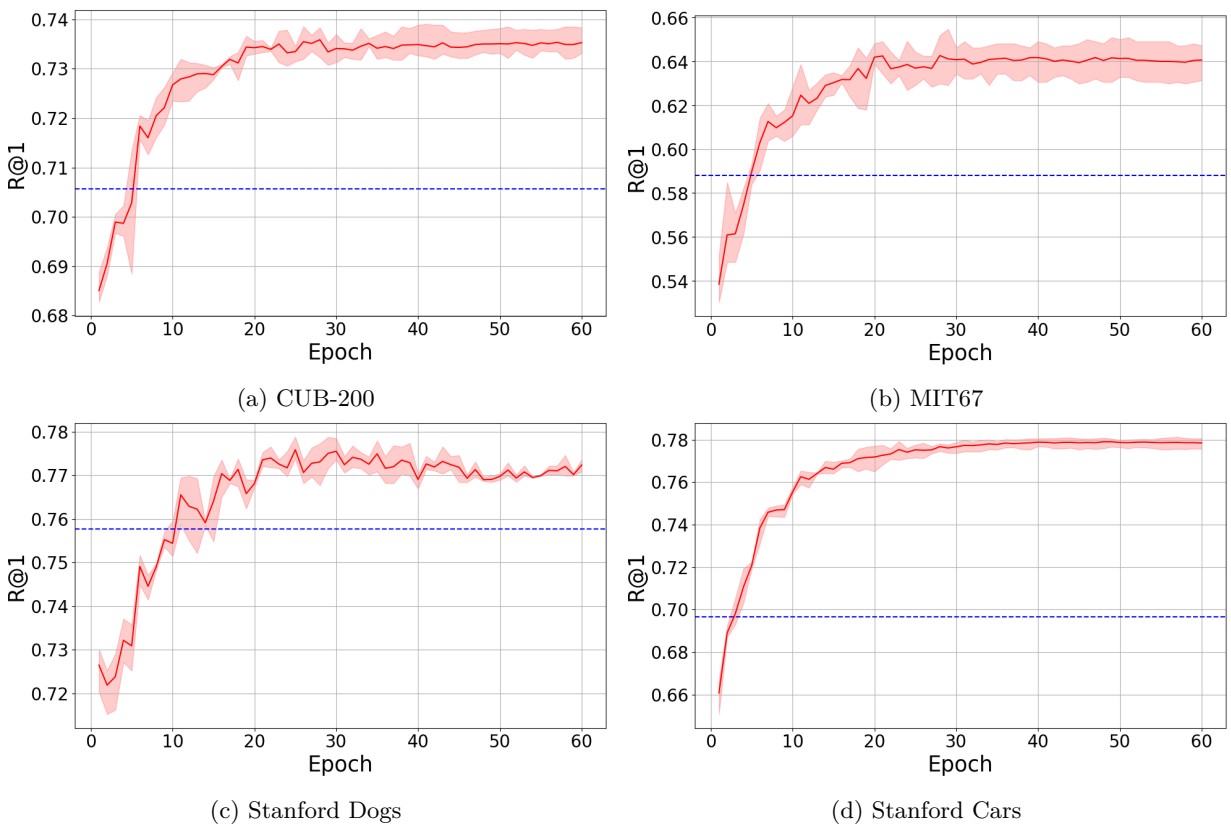

Figure 3: Performance in Recall@1 versus epoch number for fine-tuning on our *evt* loss over all runs. The blue dotted line indicates PA accuracy, i.e., the initial accuracy of the model. The accuracy drops at the beginning of training, bouncing back after a few epochs across different datasets, converging after 10-20 epochs and achieving higher performance compared to that of PA.

Table 4: Comparison of different methods across various datasets w.r.t. various metrics $\mathcal{M}_{all}$, $\mathcal{M}_o$, $\mathcal{M}_f$, and $\mathcal{M}_d$. CATEGORIZER outperforms all other methods in terms of almost all metrics. $\uparrow$ ($\downarrow$) indicates the metric should have a higher (lower) value.

| Dataset | Method | $\mathcal{M}_{all}$ | $\mathcal{M}_o \uparrow$ | $\mathcal{M}_f \downarrow$ | $\mathcal{M}_d \uparrow$ |
|---------|--------|---------------------|--------------------------|----------------------------|--------------------------|
| CUB | GM | $12.13 \pm 1.10$ | $13.52 \pm 1.51$ | $37.28 \pm 1.51$ | $10.5 \pm 1.34$ |
| | MetaGCD | $48.18 \pm 0.68$ | $52.08 \pm 1.10$ | $27.27 \pm 1.10$ | $\underline{32.42 \pm 3.06}$ |
| | PA | $56.52 \pm 0.39$ | $64.32 \pm 0.88$ | $14.39 \pm 0.88$ | $25.77 \pm 3.12$ |
| | IGCD | $\underline{56.67 \pm 0.43}$ | $\underline{66.52 \pm 1.34}$ | $\mathbf{9.55 \pm 1.34}$ | $17.28 \pm 2.98$ |
| | Happy | $46.44 \pm 0.43$ | $54.03 \pm 1.14$ | $13.27 \pm 1.14$ | $16.78 \pm 1.89$ |
| | **CATEGORIZER** | $\mathbf{61.50 \pm 0.46}$ | $\mathbf{68.48 \pm 0.97}$ | $\underline{11.72 \pm 0.97}$ | $\mathbf{33.62 \pm 3.05}$ |
| MIT | GM | $18.50 \pm 1.92$ | $19.37 \pm 1.84$ | $41.68 \pm 1.84$ | $16.93 \pm 1.97$ |
| | MetaGCD | $40.45 \pm 1.93$ | $43.40 \pm 2.04$ | $31.18 \pm 2.04$ | $\underline{29.29 \pm 2.02}$ |
| | PA | $\underline{52.84 \pm 1.56}$ | $\underline{60.19 \pm 1.92}$ | $12.45 \pm 1.92$ | $25.00 \pm 1.97$ |
| | IGCD | $44.01 \pm 1.75$ | $50.27 \pm 1.99$ | $20.25 \pm 1.99$ | $20.29 \pm 2.32$ |
| | Happy | $42.61 \pm 0.24$ | $51.42 \pm 1.34$ | $14.90 \pm 1.34$ | $9.29 \pm 1.14$ |
| | **CATEGORIZER** | $\mathbf{57.10 \pm 1.83}$ | $\mathbf{64.15 \pm 2.02}$ | $\underline{10.30 \pm 2.02}$ | $\mathbf{32.87 \pm 2.24}$ |
| Dogs | GM | $11.35 \pm 1.52$ | $13.42 \pm 1.81$ | $44.61 \pm 1.81$ | $10.23 \pm 2.73$ |
| | MetaGCD | $54.35 \pm 1.23$ | $54.9 \pm 1.45$ | $29.48 \pm 1.45$ | $34.45 \pm 4.42$ |
| | PA | $\underline{66.23 \pm 1.15}$ | $\underline{74.23 \pm 1.95}$ | $10.69 \pm 1.95$ | $34.69 \pm 4.62$ |
| | IGCD | $33.63 \pm 0.93$ | $39.15 \pm 1.38$ | $40.11 \pm 1.38$ | $11.52 \pm 2.34$ |

| Dataset | Method | $\mathcal{M}_{all}$ | $\mathcal{M}_o \uparrow$ | $\mathcal{M}_f \downarrow$ | $\mathcal{M}_d \uparrow$ |
|---|---|---|---|---|---|
| | Happy | $64.75 \pm 0.84$ | $72.06 \pm 1.22$ | $10.66 \pm 1.22$ | $35.06 \pm 1.99$ |
| | **CATEGORIZER** | $\mathbf{68.46 \pm 1.05}$ | $\mathbf{76.42 \pm 1.79}$ | $\mathbf{6.98 \pm 1.79}$ | $\mathbf{37.09 \pm 4.50}$ |
| CIFAR10 | GM | $8.21 \pm 1.49$ | $20.32 \pm 1.36$ | $65.37 \pm 1.36$ | $4.23 \pm 2.39$ |
| | MetaGCD | $9.55 \pm 1.33$ | $23.86 \pm 1.73$ | $49.36 \pm 1.73$ | $8.47 \pm 2.62$ |
| | PA | $64.39 \pm 1.37$ | $79.07 \pm 1.89$ | $17.14 \pm 1.89$ | $5.65 \pm 2.68$ |
| | IGCD | $20.13 \pm 0.67$ | $20.46 \pm 2.48$ | $45.44 \pm 2.48$ | $8.80 \pm 1.94$ |
| | Happy | $53.14 \pm 1.23$ | $54.15 \pm 1.18$ | $16.96 \pm 1.18$ | $9.20 \pm 1.63$ |
| | CATEGORIZER | $\mathbf{73.02 \pm 1.35}$ | $\mathbf{87.90 \pm 1.54}$ | $\mathbf{3.88 \pm 1.54}$ | $\mathbf{11.73 \pm 2.32}$ |
| CIFAR100 | GM | $8.32 \pm 1.39$ | $9.51 \pm 1.71$ | $69.94 \pm 1.71$ | $5.23 \pm 2.46$ |
| | MetaGCD | $34.14 \pm 1.47$ | $35.60 \pm 1.85$ | $49.36 \pm 1.85$ | $\mathbf{28.30 \pm 2.71}$ |
| | PA | $61.30 \pm 1.19$ | $74.45 \pm 1.89$ | $6.46 \pm 1.89$ | $8.70 \pm 3.63$ |
| | IGCD | $10.15 \pm 0.67$ | $10.86 \pm 2.48$ | $71.37 \pm 2.48$ | $7.30 \pm 1.94$ |
| | Happy | $44.70 \pm 1.27$ | $25.09 \pm 1.12$ | $14.75 \pm 1.12$ | $26.70 \pm 1.11$ |
| | CATEGORIZER | $\mathbf{64.01 \pm 1.35}$ | $\mathbf{76.95 \pm 1.54}$ | $\mathbf{4.51 \pm 1.63}$ | $14.72 \pm 2.51$ |
| AIRCRAFT | GM | $14.82 \pm 1.42$ | $17.45 \pm 1.21$ | $46.87 \pm 1.21$ | $11.23 \pm 2.35$ |
| | MetaGCD | $\mathbf{45.65 \pm 1.36}$ | $45.65 \pm 1.74$ | $26.35 \pm 1.74$ | $\mathbf{39.28 \pm 2.63}$ |
| | PA | $37.62 \pm 1.15$ | $42.46 \pm 1.76$ | $18.34 \pm 1.76$ | $18.29 \pm 3.21$ |
| | IGCD | $42.11 \pm 0.57$ | $47.62 \pm 2.56$ | $29.35 \pm 2.56$ | $20.09 \pm 1.73$ |
| | Happy | $24.36 \pm 0.67$ | $28.39 \pm 1.13$ | $15.98 \pm 1.13$ | $8.25 \pm 1.94$ |
| | CATEGORIZER | $45.24 \pm 1.29$ | $\mathbf{49.74 \pm 1.38}$ | $13.46 \pm 1.38$ | $27.29 \pm 2.57$ |
| SVHN | GM | $10.81 \pm 1.31$ | $16.32 \pm 1.32$ | $46.87 \pm 1.32$ | $10.23 \pm 1.34$ |
| | MetaGCD | $20.76 \pm 1.23$ | $19.11 \pm 1.39$ | $28.31 \pm 1.39$ | $25.20 \pm 1.95$ |
| | PA | $73.28 \pm 1.27$ | $82.66 \pm 1.25$ | $13.88 \pm 1.25$ | $45.23 \pm 2.28$ |
| | IGCD | $11.79 \pm 1.32$ | $9.85 \pm 1.36$ | $86.87 \pm 1.36$ | $19.52 \pm 2.21$ |
| | Happy | $20.13 \pm 0.67$ | $20.46 \pm 2.48$ | $45.44 \pm 2.48$ | $8.80 \pm 1.94$ |
| | CATEGORIZER | $\mathbf{75.21 \pm 1.39}$ | $\mathbf{84.32 \pm 1.24}$ | $\mathbf{11.46 \pm 1.24}$ | $\mathbf{52.01 \pm 2.31}$ |
| mnist | GM | $58.72 \pm 1.24$ | $60.27 \pm 1.21$ | $35.23 \pm 1.41$ | $40.27 \pm 2.11$ |
| | MetaGCD | $73.20 \pm 1.03$ | $76.90 \pm 1.15$ | $18.24 \pm 1.24$ | $45.14 \pm 1.96$ |
| | PA | $77.48 \pm 1.27$ | $85.65 \pm 1.25$ | $13.47 \pm 1.25$ | $45.32 \pm 2.17$ |
| | IGCD | $74.12 \pm 1.45$ | $83.43 \pm 1.28$ | $15.92 \pm 1.28$ | $36.88 \pm 2.36$ |
| | Happy | $60.99 \pm 1.46$ | $65.43 \pm 1.22$ | $20.62 \pm 1.22$ | $43.33 \pm 2.73$ |
| | CATEGORIZER | $\mathbf{79.90 \pm 1.29}$ | $\mathbf{87.82 \pm 1.19}$ | $\mathbf{11.23 \pm 1.19}$ | $\mathbf{49.15 \pm 2.28}$ |
| Fashion-mnist | GM | $28.72 \pm 1.32$ | $32.16 \pm 1.11$ | $65.24 \pm 1.11$ | $4.17 \pm 2.22$ |
| | MetaGCD | $52.76 \pm 1.27$ | $61.91 \pm 1.14$ | $35.69 \pm 1.14$ | $25.16 \pm 1.86$ |
| | PA | $50.13 \pm 1.18$ | $60.61 \pm 1.22$ | $37.21 \pm 1.22$ | $7.50 \pm 2.25$ |
| | IGCD | $48.29 \pm 1.31$ | $57.51 \pm 1.17$ | $40.34 \pm 1.17$ | $11.40 \pm 2.21$ |
| | Happy | $51.78 \pm 1.37$ | $58.86 \pm 1.98$ | $27.35 \pm 1.98$ | $8.80 \pm 1.94$ |
| | CATEGORIZER | $\mathbf{61.02 \pm 1.29}$ | $\mathbf{65.82 \pm 1.13}$ | $\mathbf{26.84 \pm 1.13}$ | $\mathbf{27.87 \pm 2.17}$ |
| Cars | GM | $23.52 \pm 2.32$ | $28.24 \pm 2.67$ | $43.26 \pm 2.67$ | $17.23 \pm 1.96$ |
| | MetaGCD | $47.17 \pm 2.23$ | $52.99 \pm 2.34$ | $23.69 \pm 2.34$ | $24.71 \pm 4.53$ |
| | PA | $40.46 \pm 2.09$ | $44.35 \pm 2.55$ | $24.30 \pm 2.55$ | $25.44 \pm 3.62$ |
| | IGCD | $48.60 \pm 1.88$ | $57.82 \pm 2.57$ | $19.33 \pm 2.57$ | $12.63 \pm 2.75$ |
| | Happy | $15.06 \pm 0.38$ | $16.54 \pm 1.27$ | $19.17 \pm 1.27$ | $9.38 \pm 1.28$ |
| | **CATEGORIZER** | $\mathbf{52.97 \pm 2.13}$ | $\mathbf{59.34 \pm 2.46}$ | $17.75 \pm 2.46$ | $\mathbf{33.29 \pm 4.68}$ |

## 5.3 Comparison with State-of-the-art Methods

We conducted a series of experiments to evaluate CATEGORIZER against state-of-the-art (SOTA) approaches in the CGCD setting. For comparisons, we used the IGCD Zhao & Mac Aodha (2023), GM Zhang et al. (2022), PA Kim et al. (2023), MetaGCD Wu et al. (2023), and Happy Ma et al. (2024), which are very recent works and operate in the same context. Following the experimental setup from Kim et al. (2023), we used 80% of the classes in the initial stage and introduced the remaining 20% in the continual learning stage. To better reflect real-world scenarios, where incoming data may include both known and novel classes, only

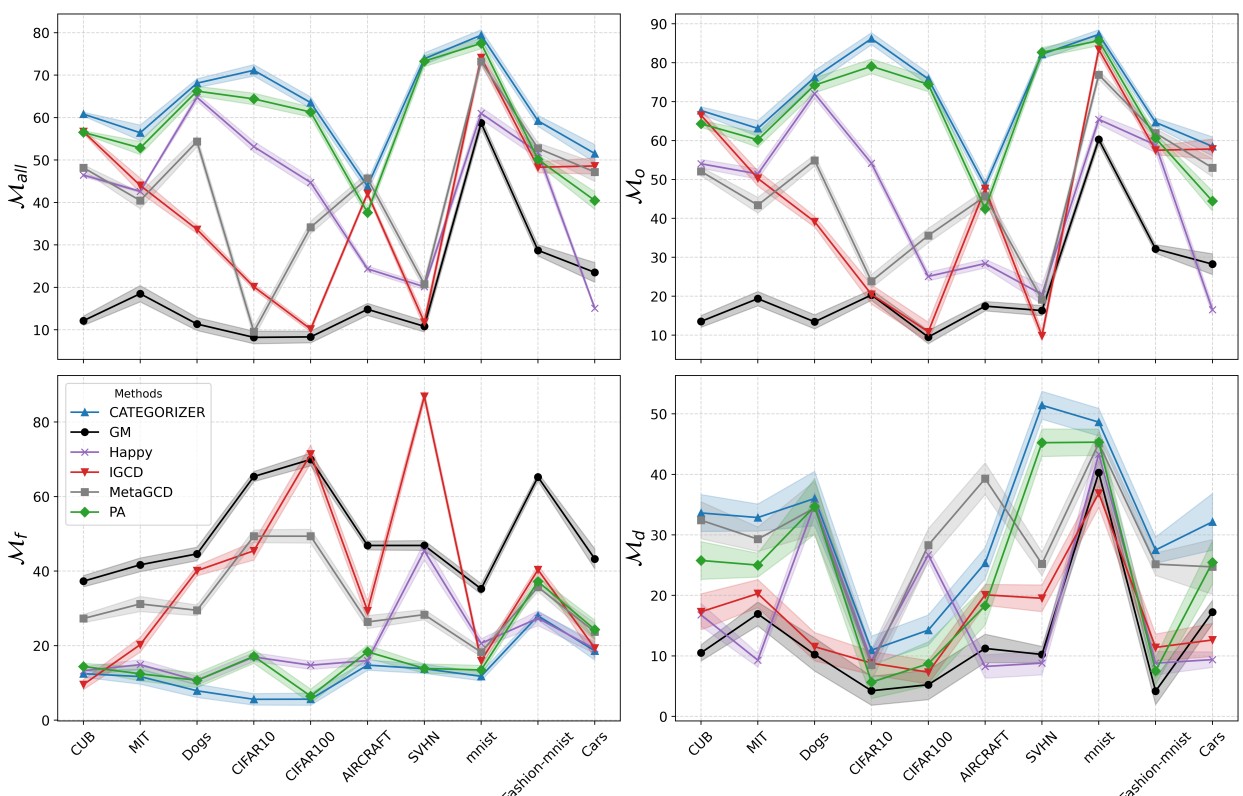

Figure 4: Performance comparison of different methods across multiple datasets using four metrics: $\mathcal{M}_{all}$, $\mathcal{M}_o$, $\mathcal{M}_f$, and $\mathcal{M}_d$. Each subplot corresponds to one metric, where the x-axis represents datasets and the y-axis indicates the metric score. Lines represent the average performance of each method, with shaded regions showing the standard deviation across runs. CATEGORIZER consistently achieves top performance across most metrics and datasets.

80% of the data was utilized during the initial stage, while the remaining 20% of samples from known classes were mixed with novel samples in the continual learning stage.

For our experiments, we used fine-grained datasets that closely resemble real-world scenarios and some general datasets: birds Wah et al. (2011), indoor scenes Quattoni & Torralba (2009), cars Krause et al. (2013), dogs Khosla et al. (2011), CIFAR10 Krizhevsky et al. (2009), CIFAR100 Krizhevsky et al. (2009), SVHN Netzer et al. (2011), mnist Deng (2012), Fashion-mnist Xiao et al. (2017), and aircraft Maji et al. (2013). According to the experimental setup, the initial and continual stage class splits were as follows: birds (160/40), cars (156/40), indoor scenes (53/14), dogs (96/24), CIFAR10 (8/2), CIFAR100 (80/20), SVHN (8/2), MNIST (8/2), Fashion-mnist (8/2), and aircraft (80/20). It is important to highlight that our proposed framework is not limited to a specific data type, such as images. It can be applied to any type of data, as long as a suitable backbone network is employed for feature extraction. However, since prior approaches in the CGCD setting have utilized image datasets for their results, we opted to use them as well.

In addition to accuracy, we evaluated the recall at K metric for more fine-grained datasets such as CUB (since the difference in performance on other datasets is negligible), to measure the learned representation of different methods in the initial stage as it is essential for novelty detection and overall performance. Results for recall at K (K = 1, 2, 4, and 8) are reported in Table 3. Our proposed *evt* loss significantly improves the performance, particularly on the cars dataset, compared to plain PA as shown in Figure 3, which reports the Recall@1 performance versus epoch number. Overall, our proposed approach surpasses other methods in terms of representation learning.

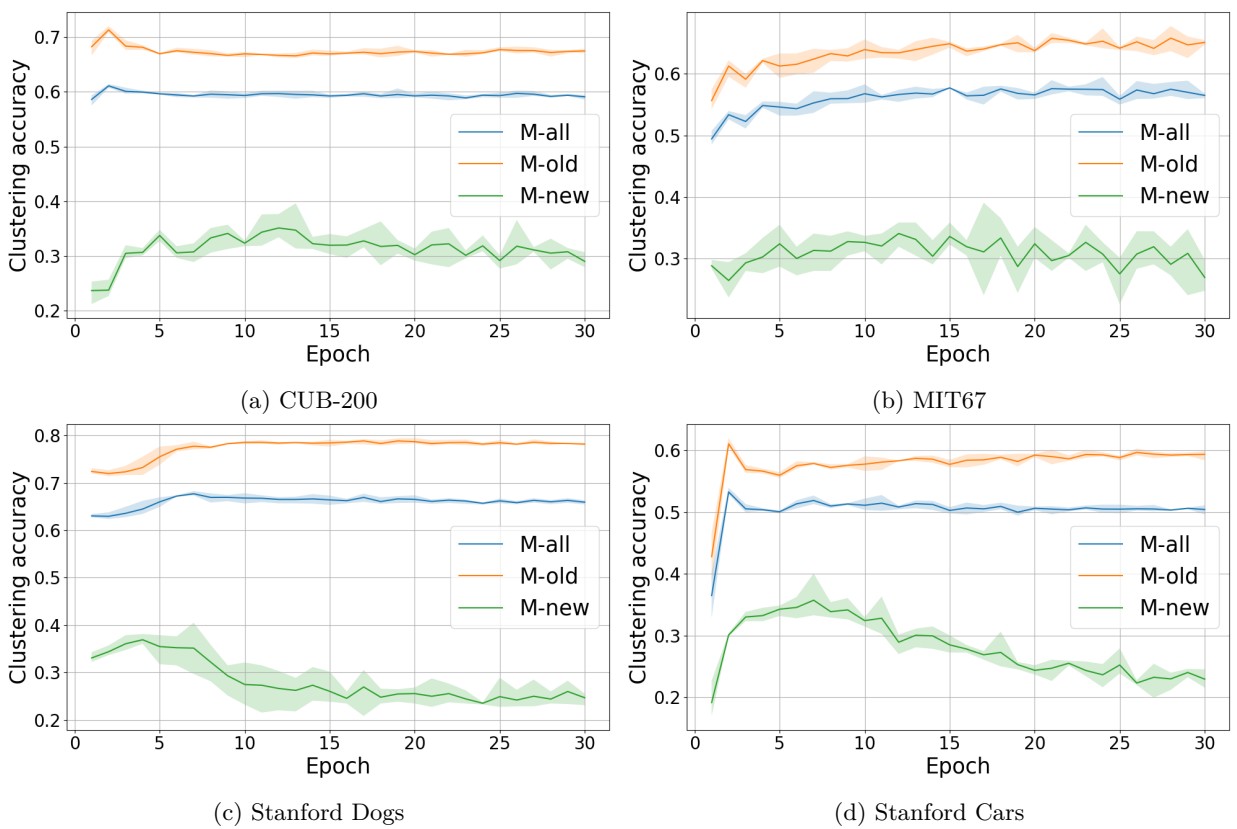

Figure 5: Clustering accuracy versus epoch number in the continual learning stage over all runs. In the Dogs and Cars datasets, training for longer epochs has led to feature collapse of newly discovered classes, while in CUB and MIT datasets, the accuracy has not changed after certain epochs. Based on this observation we limit the number of epochs of the model training in the continual learning stage.

Table 4 compares different methods during the continual learning stage. Figure 4 illustrates the comparative performance of state-of-the-art methods and CATEGORIZER across multiple datasets. CATEGORIZER consistently achieves superior or comparable results compared to other methods across different datasets and evaluation metrics, including novel class discovery ($M_d$), retention of prior knowledge ($M_o$ and $M_f$), and overall performance ($M_{all}$). On some datasets like birds, IGCD exhibited better forgetting performance, likely because its initial accuracy was lower than ours. GM performs poorly compared to others, which is due to the fact that it requires the ratio of novel category samples on the new dataset. MetaGCD showed good performance in terms of the $M_d$ metric, demonstrating its ability to learn novel classes effectively, but struggled to retain performance on previously known classes. PA performed well on most datasets, especially dogs, but showed weaker results for the cars dataset, which can be explained by its relatively poor learned representation, as reported in Table 3. Happy shows strong performance on the $\mathcal{M}_d$ metric for the CIFAR100 and Dogs datasets, but does not exhibit notably better results across the other metrics.

Table 5: Ablation study on the effect of *evt* loss and model reduction (RED).

|  | $\mathcal{L}_{evt}$ | **Red** | $\mathcal{M}_{all}$ | $\mathcal{M}_o\uparrow$ | $\mathcal{M}_f\downarrow$ | $\mathcal{M}_d\uparrow$ |
|---|---|---|---|---|---|---|
| | ✗ | ✗ | 57.03 | 65.65 | 12.70 | 23.00 |
| CUB | ✓ | ✗ | 59.44 | 66.05 | 14.15 | 33.36 |
| | ✗ | ✓ | 59.39 | 67.78 | 10.82 | 26.28 |
| | ✓ | ✓ | 61.50 | 68.48 | 11.72 | 33.62 |
| | ✗ | ✗ | 50.93 | 58.74 | 13.52 | 21.36 |
| MIT | ✓ | ✗ | 55.82 | 63.87 | 10.94 | 25.71 |
| | ✗ | ✓ | 55.72 | 64.09 | 10.72 | 25.00 |

| $\mathcal{L}_{evt}$ | **Red** | $\mathcal{M}_{all}$ | $\mathcal{M}_o \uparrow$ | $\mathcal{M}_f \downarrow$ | $\mathcal{M}_d \uparrow$ | |
|---|---|---|---|---|---|---|
| | ✓ | ✓ | 57.10 | 64.15 | 10.30 | 32.87 |
| Dogs | ✗ | ✗ | 62.35 | 73.12 | 11.68 | 23.81 |
| | ✓ | ✗ | 63.36 | 72.07 | 12.06 | 29.05 |
| | ✗ | ✓ | 66.40 | 77.89 | 6.75 | 22.09 |
| | ✓ | ✓ | 68.46 | 76.42 | 6.98 | 37.09 |
| CIFAR10 | ✗ | ✗ | 60.15 | 75.47 | 16.31 | 20.04 |
| | ✓ | ✗ | 66.28 | 80.35 | 11.43 | 24.54 |
| | ✗ | ✓ | 69.10 | 83.00 | 8.78 | 27.85 |
| | ✓ | ✓ | 73.02 | 87.90 | 3.88 | 11.73 |
| CIFAR100 | ✗ | ✗ | 55.03 | 55.03 | 14.12 | 18.64 |
| | ✓ | ✗ | 60.00 | 60.00 | 9.15 | 22.08 |
| | ✗ | ✓ | 62.40 | 62.40 | 6.75 | 24.12 |
| | ✓ | ✓ | 64.01 | 76.95 | 4.51 | 14.72 |
| AIRCRAFT | ✗ | ✗ | 41.92 | 47.65 | 15.55 | 20.45 |
| | ✓ | ✗ | 44.32 | 49.90 | 13.30 | 22.78 |
| | ✗ | ✓ | 43.00 | 47.23 | 15.97 | 21.36 |
| | ✓ | ✓ | 45.24 | 49.74 | 13.46 | 27.29 |
| SVHN | ✗ | ✗ | 68.10 | 72.85 | 23.11 | 33.25 |
| | ✓ | ✗ | 72.10 | 77.50 | 18.46 | 36.25 |
| | ✗ | ✓ | 73.40 | 80.20 | 15.76 | 38.10 |
| | ✓ | ✓ | 75.21 | 84.32 | 11.64 | 52.01 |
| MNIST | ✗ | ✗ | 60.50 | 71.10 | 27.95 | 35.50 |
| | ✓ | ✗ | 70.50 | 80.25 | 18.80 | 40.20 |
| | ✗ | ✓ | 74.15 | 82.95 | 16.10 | 41.40 |
| | ✓ | ✓ | 79.90 | 87.82 | 11.23 | 49.15 |
| FASHION-MNIST | ✗ | ✗ | 55.80 | 61.80 | 30.86 | 20.00 |
| | ✓ | ✗ | 56.90 | 62.90 | 29.76 | 21.80 |
| | ✗ | ✓ | 58.50 | 63.50 | 29.16 | 23.50 |
| | ✓ | ✓ | 61.02 | 65.82 | 26.84 | 27.87 |
| Cars | ✗ | ✗ | 45.92 | 52.37 | 24, 72 | 21.05 |
| | ✓ | ✗ | 47.72 | 52.80 | 24.29 | 29.06 |
| | ✗ | ✓ | 48.46 | 55.50 | 21.59 | 22.18 |
| | ✓ | ✓ | 52.97 | 59.34 | 17.75 | 33.29 |

Table 6: Effect of the model reduction on the estimated number of categories. With the model reduction, the estimations are significantly closer to the actual number of categories

| | # OF CATEGORIES | W/O REDUCTION | WITH REDUCTION |
|---|---|---|---|
| CUB | 200 | 285 | 231 |
| MIT | 67 | 134 | 81 |
| DOGS | 120 | 239 | 141 |
| CIFAR10 | 10 | 120 | 37 |
| CIFAR100 | 100 | 230 | 120 |
| AIRCRAFT | 100 | 110 | 19 |
| SVHN | 10 | 180 | 21 |
| MNIST | 10 | 175 | 20 |
| FASHION-MNIST | 10 | 185 | 24 |
| CARS | 196 | 354 | 222 |

## 5.4 Ablation Study

We study the effect of our *evt* loss and cluster reduction on the performance of the proposed method, which are reported in Table 5.

**Effectiveness of the** *evt* **loss**: As reported in Table 5, the proposed *evt* loss enhances performance in terms of most metrics, with a particularly notable improvement in terms of $M_d$, highlighting its significance for the novelty detection module. However, on certain datasets, such as Stanford Dogs, the $M_o$ metric shows reduced performance when using this loss, in exchange for a higher $M_d$. This indicates a suboptimal trade-off between plasticity and stability, which is mitigated by integrating the *evt* loss with model-reduction techniques. Additionally, the method exhibits increased forgetting on some datasets due to the *evt* loss boosting the initial accuracy, making subsequent accuracy drops more pronounced.

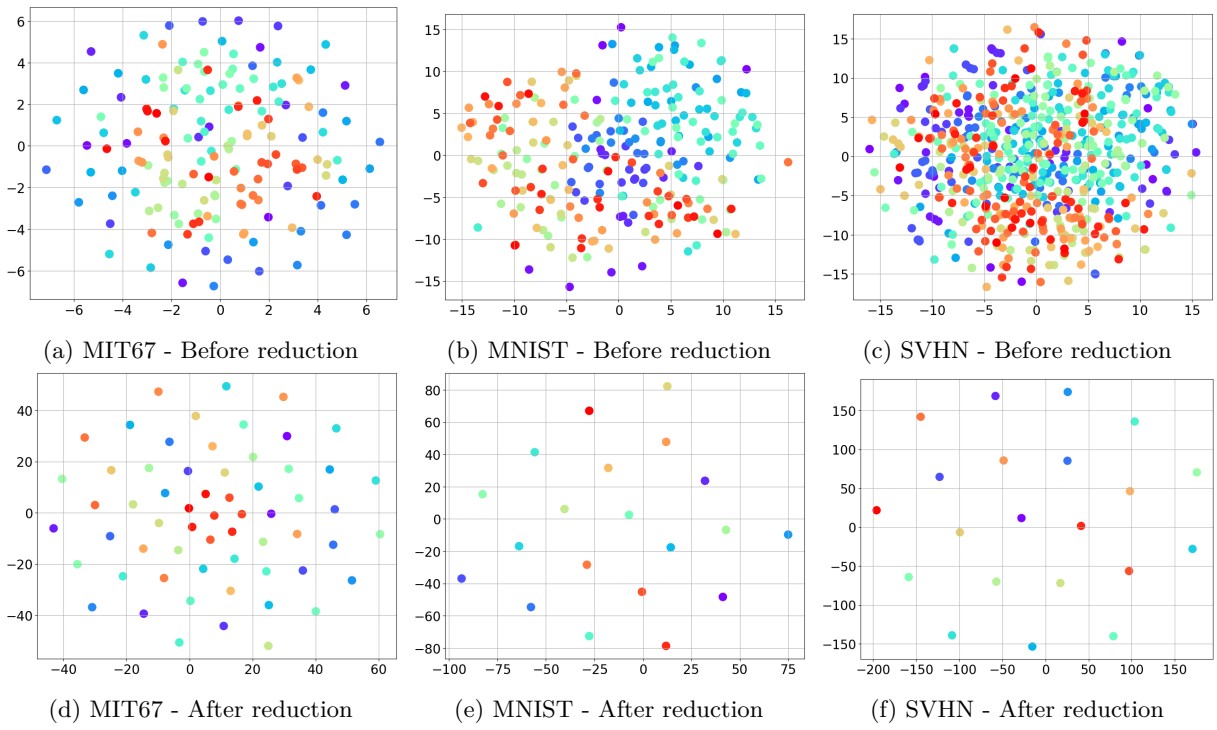

(a) MIT67 - Before reduction  (b) MNIST - Before reduction  (c) SVHN - Before reduction

(d) MIT67 - After reduction  (e) MNIST - After reduction  (f) SVHN - After reduction

Figure 6: Distribution of proxies in the feature space before and after reduction on MIT67, MNIST, and SVHN. The reduction procedure eliminates redundant proxies, yielding a closer approximation of the true number of categories, which enhances accuracy and reduces prediction time.

Table 7: Comparison of novelty detection module of CATEGORIZER and PA. The results show improvement across all the datasets.

|  | PA | CATEGORIZER |
|---|---|---|
| CUB | 59.77 | **70.57** |
| MIT | 60.30 | **67.20** |
| Dogs | 65.26 | **68.74** |
| CIFAR10 | 63.20 | **70.00** |
| CIFAR100 | 66.10 | **69.10** |
| AIRCRAFT | 65.23 | **70.20** |
| SVHN | 63.86 | **69.50** |
| MNIST | 62.11 | **68.30** |
| FASHION-MNIST | 65.32 | **70.10** |
| CARS | 69.10 | **71.66** |

**Effectiveness of model reduction**: Per Table 5, model reduction introduced in Section 4.2.3 could improve the proposed method in terms of most metrics, particularly the $M_o$ metric. We believe this improvement is

due to the fact that having redundant proxies for the novel categories significantly degrades the performance of old categories. Removing these redundant proxies helps to maintain the model's stability. The combination of the *evt* loss and model reduction has achieved the most optimal trade-off between maintaining accuracy on the old categories ($M_o$) and effective discovery and learning novel categories ($M_d$). Table 6 shows the effect of the model reduction module on the estimated number of categories in the discovery phase. Upon performing model reduction, the estimated number of categories is significantly closer to the real number of categories. Figure 6 shows the removal of redundant proxies in the feature space, visualized with t-SNE Maaten & Hinton (2008), resulting in a distribution that more accurately reflects the true number of categories.

**Effectiveness on the novelty detection module**: Table 7 shows the improvement in the accuracy of the novelty detection module of CATEGORIZER compared to the original PA method Kim et al. (2023). CATEGORIZER has consistently improved the novelty detection accuracy method across different datasets.

Table 8: Performance comparison using Gaussian and WEIBULL distributions across datasets.

| Dataset | Distribution | $M_{all}$ | $M_{old}$ | $M_f$ | $M_d$ |
|---|---|---|---|---|---|
| Dogs | Gaussian | 68.10 | 76.24 | 7.89 | 36.01 |
|  | WEIBULL | **68.46** | **76.42** | **6.98** | **37.09** |
| Cub | Gaussian | 60.82 | 67.72 | 12.48 | 33.62 |
|  | WEIBULL | **61.50** | **68.48** | **11.72** | **33.61** |
| Cars | Gaussian | 51.52 | 58.50 | 18.59 | 32.15 |
|  | WEIBULL | **52.97** | **59.34** | **17.75** | **33.29** |
| Air | Gaussian | 43.80 | 48.42 | 14.78 | 25.34 |
|  | WEIBULL | **45.24** | **49.74** | **13.46** | **27.29** |
| Mit | Gaussian | 56.42 | 63.11 | 11.70 | 32.86 |
|  | WEIBULL | **57.10** | **64.15** | **10.30** | **32.87** |
| Cifar10 | Gaussian | 71.13 | 86.18 | 5.60 | 10.95 |
|  | WEIBULL | **73.02** | **87.90** | **3.88** | **11.73** |
| SVHN | Gaussian | 73.86 | 82.16 | 13.80 | 51.42 |
|  | WEIBULL | **75.21** | **84.32** | **11.64** | **52.01** |
| Cifar100 | Gaussian | 63.51 | 75.82 | 5.64 | 14.25 |
|  | WEIBULL | **64.01** | **76.95** | **4.51** | **14.72** |
| MNIST | Gaussian | 79.40 | 87.28 | 11.77 | 48.63 |
|  | WEIBULL | **79.90** | **87.82** | **11.23** | **49.15** |
| Fashion-mnist | Gaussian | 59.23 | 64.66 | 28.00 | 27.50 |
|  | WEIBULL | **61.02** | **65.82** | **26.84** | **27.87** |

**Effectiveness of EVT-based feature sampling**: Table 8 presents a comparison between using the Weibull distribution from EVT and using a Gaussian distribution (used in Kim et al. (2023)) in the feature replay phase. Employing the Weibull distribution leads to improved performance across all metrics and for all datasets.

# 6    Conclusion

In this paper, we proposed a novel method, called CATEGORIZER, to handle the continual generalized category discovery problem. CATEGORIZER combines proxy anchors and extreme value theory to define decision boundaries around each proxy. We proposed a novel loss, called the *evt* loss which, enhances the learned representation compared to the plain proxy anchors and outperforms deep metric learning loss used in similar SOTA methods in the CGCD scenario. In addition, we demonstrated that employing a Weibull distribution instead of a Gaussian distribution in the feature-replay phase consistently leads to improved performance. Furthermore, we mitigated the problem of over-estimating the number of novel categories in

the discovery phase by resorting to a novel method, which is based on EVT. CATEGORIZER outperforms state-of-the-art methods in the CGCD scenario across different datasets. In future work, we plan to integrate the *evt* loss during the continual learning stage of the framework. In addition, we will investigate other clustering methods to be used in the discovery step.

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

## A  Appendix

Table 9 presents the number of categories estimated by different clustering methods. Our evaluation across multiple clustering algorithms confirms that the overestimation issue persists regardless of the method used. Affinity Propagation Frey & Dueck (2007) and Hierarchical Clustering Ward Jr (1963) exhibit similar and stable behavior, whereas Birch Zhang et al. (1996) consistently overestimates the number of clusters. In contrast, density-based methods (DBSCAN Ester et al. (1996), HDBSCAN McInnes et al. (2017), and OPTICS Ankerst et al. (1999)) are highly sensitive to hyperparameters such as min samples, leading to unstable results and rendering them unsuitable for CGCD's continually changing environment.

Table 9: Comparison of the number of clusters estimated by different clustering methods across datasets.

| Dataset | Actual | Affinity | Birch | Hierarchical |
|---------|--------|----------|-------|--------------|
| Dogs | 120 | 229 | 437 | 229 |
| Cub | 200 | 285 | 496 | 285 |
| Cars | 196 | 354 | 250 | 355 |
| Air | 100 | 110 | 115 | 110 |
| Mit | 67 | 134 | 140 | 134 |
| Cifar10 | 10 | 120 | 236 | 120 |
| SVHN | 10 | 180 | 210 | 179 |
| Cifar100 | 100 | 230 | 811 | 232 |
| MNIST | 10 | 175 | 130 | 175 |
| Fashion-MNIST | 10 | 185 | 232 | 185 |

