# OpenReview forum: "Proxy-Anchor and EVT-Driven Continual Learning Method for Generalized Category Discovery"
_TMLR — Accepted by TMLR_

### Review · Reviewer_ZHgS · 2025-11-25

**Summary Of Contributions:**

1. The paper addresses an important and challenging problem in machine learning.
2. It introduces a new approach, CATEGORIZER, which achieves superior performance compared to existing state-of-the-art (SOTA) methods.
3. The proposed method effectively mitigates the tendency to overestimate the number of novel categories during the discovery phase.

**Audience:**

Yes

**Audience Explanation:**

I believe that, given the challenging and important problem setting, along with the clear and comprehensive methodology and evaluations, this work should be valuable to at least a portion of TMLR’s audience.

**Broader Impact Concerns:**

I think the main limitation of this paper is that, although Continual GCD is an important setting, the specific formulation presented in Section 3 is rather restrictive. In real-world scenarios, it is often unrealistic to assume that the model first encounters labeled data and then unlabeled data sequentially, making this setup somewhat limited in practical applicability.

**Claims And Evidence:**

Yes

**Claims Explanation:**

1. The problem setting in the paper is well formulated and the proposed method is well illustrated.
2. Despite that introducing EVT or Proxy loss into this Continual GCD setting is not entirely new but the pipeline is quite intuitive.
3. The evaluation is also abundant and convincing from my perspective.

**Requested Changes:**

The overall writing is clear and easy to understand. Some captions and characters of Figure 1 is not clear enough.

---

> ### Author Response · Authors · 2025-12-08
> **Addressed Comments**
>
> Thanks for your thoughtful and insightful comments.
>
> We’ll make sure to increase the font size in Figure 1 to make it more readable.
>
> Regarding the broader impact comment, during the initial stage, the user has the time and resources to gather labeled data to initialize the model. Although collecting a labeled dataset is always a challenging task, there are various ways to achieve this, such as manual labeling, using semi-supervised approaches to propagate labels, and other similar methods. However, once the model is deployed in the real world, the user no longer has the luxury of providing clean labeled data, and the model must operate in an almost entirely unsupervised manner.

---

### Review · Reviewer_GL8n · 2025-11-26

**Summary Of Contributions:**

Summary
The paper introduces "CATEGORIZER", a framework designed for Continual Generalized Category Discovery (CGCD) that integrates Extreme Value Theory (EVT) with Proxy Anchor learning. By leveraging Weibull distributions to model the decision boundaries of proxies, this paper proposes a novel evt loss to enhance representation quality and a Probability of Inclusion function for effective novelty detection. To address the over-clustering issue inherent in Affinity Propagation during the discovery phase, an EVT-driven Model Reduction strategy is employed to merge redundant proxies via greedy set cover optimization. Experiments on benchmarks like CUB-200 demonstrate that the proposed method achieves superior clustering accuracy and more accurate category count estimation compared to state-of-the-art methods.

**Audience:**

Yes

**Audience Explanation:**

The paper proposes a novel Continual Learning Method for Generalized Category Discovery which may inspire the community.

**Broader Impact Concerns:**

No further broader impact concerns.

**Claims And Evidence:**

Yes

**Claims Explanation:**

The paper has the following three Strengths:
1.The paper proposes an innovative approach by incorporating Extreme Value Theory directly into the training objective via the evt loss. This effectively shapes the embedding space to better fit Weibull distributions, improving the separation between known and unknown categories during the initial stage.
2.This paper presents a comprehensive framework that systematically addresses the interconnected sub-tasks of novelty detection, category discovery, and continual learning. This design explicitly targets the challenge of class count over-estimation, providing a unified solution for the CGCD problem.
3.The Model Reduction module offers a structured mechanism based on Set Cover optimization to refine the granularity of discovered concepts. This effectively mitigates the over-clustering limitations of affinity-based methods, as demonstrated by the improved proxy distributions in the visualization.

**Requested Changes:**

The following concerns may impact the final decision of this paper.
1. The proposed evt loss (Eq. 3) depends on Weibull parameters ($\lambda_p, \kappa_p$) derived from the network's feature distribution, but Algorithm 1 indicates these are estimated only once in Step 2 and treated as fixed constants throughout the fine-tuning loop. This results in the model being optimized against outdated statistical boundaries that do not reflect the evolving feature space. The paper lacks theoretical proof or experimental evidence (e.g., loss curves) to demonstrate the convergence of this specific scheme. Furthermore, applying this evt loss in the continual learning stage would likely amplify these optimization inconsistencies.
2. In Section 4.2.2, the paper employs Affinity Propagation for novel class discovery, explicitly noting that it results in over-clustering. To mitigate this, a Model Reduction module is introduced. However, the paper does not include comparisons with other robust non-parametric clustering algorithms that might inherently avoid such over-segmentation. Without these comparisons, it is difficult to assess whether the proposed reduction module is essential or if the issue could be addressed by employing a different clustering method.
3. Although the Class Incremental Learning stage is a pivotal component of the framework, it lacks technical innovation by relying on standard feature distillation and replay techniques without modification. Furthermore, the paper fails to verify the actual effectiveness of these adopted components.
4. Table 6 shows that the estimated category counts still deviate notably from the ground truth. For example, on the Aircraft dataset, the method estimates only 19 categories compared to the actual 100, indicating that the reduction strategy fails to accurately approximate the true class distribution.

---

> ### Author Response · Authors · 2025-12-08
> **Addressed Issues**
>
> Thank you for your valuable feedback. We have addressed your comments as outlined below:
>
> 1. I appreciate your thoughtful comment. The Weibull parameters are recalculated after each stage to match the evolving embedding space, as shown in Algorithm 2, line 20 (I'll make sure to make this more clear in the revised version). To reduce drastic parameter shifts—especially given the reliability of the initial stage with more labeled data—we revise the continual learning stage to sample replayed features from the Weibull distribution instead of a Normal one. This preserves the initial parameters, improves performance, and also addresses the reviewer’s concern about limited novelty in the continual learning component (more details in the response to comment 3).
>
>
> 2. Thank you for your insightful comment. We evaluated multiple clustering methods and confirmed that the over-estimation issue persists across algorithms. Affinity Propagation and Hierarchical Clustering perform similarly and remain stable, while Birch consistently over-estimates the number of clusters. Density-based methods (DBSCAN, HDBSCAN, OPTICS) were highly sensitive to hyperparameters like min_samples, resulting in instability and making them unsuitable for CGCD’s continually changing environment. A comparison—including the actual number of categories—is provided below.
>
>
>
> | Dataset   | Actual | Affinity | Birch | Hierarchical |
> |-----------|--------|----------|-------|--------------|
> | Dogs      | 120    | 229      | 437   | 229          |
> | Cub       | 200    | 285      | 496   | 285          |
> | Cars      | 196    | 354      | 250   | 355          |
> | Air       | 100    | 110      | 115   | 110          |
> | mit       | 67     | 134      | 140   | 134          |
> | cifar10   | 10     | 120      | 236   | 120          |
> | svhn      | 10     | 180      | 210   | 179          |
> | cifar100  | 100    | 230      | 811   | 232          |
> | mnist     | 10     | 175      | 130   | 175          |
> | fashion   | 10     | 185      | 232   | 185          |
>
>
>
>
> 3. Thanks for this thoughtful comment. To improve our work and novelty of the continual learning stage, we introduce a new feature-generation strategy for the replay mechanism that improves performance across all datasets and metrics. Instead of sampling from a Gaussian distribution, we use the EVM’s class-specific Weibull parameters to generate exemplars that better reflect the true feature distribution. For each proxy, cosine distances are sampled from its Weibull distribution, and unit-norm exemplars are placed on the hypersphere via spherical interpolation. This respects the geometry of cosine space, produces more realistic replay features, and balances categories when determining the number of generated samples. Using these generated features, we define the feature replay loss as follows:
>
> \begin{equation} \mathcal{L}{fr}^t(\tilde{Z}) = \mathcal{L}{pa}^t(\tilde{Z}), \quad\tilde{Z} = \{ \tilde{z}_c \mid \tilde{z}_c = \text{SphInterp}(p^{t-1}_c, d_c), \ d_c \sim \text{Weibull}(\kappa_c, \lambda_c) \}\end{equation}
>
> This approach has improved the results as you can see as follows:
>
>
>
> | Dataset   | Distribution  | M_all  | M_old  | M_f   | M_d   |
> |-----------|---------------|--------|--------|-------|-------|
> | Dogs      | Normal Dist   | 68.10  | 76.24  | 7.89  | 36.01 |
> |           | Weibull Dist  | 68.46  | 76.42  | 6.98  | 37.09 |
> | Cub       | Normal Dist   | 60.82  | 67.72  | 12.48 | 33.62 |
> |           | Weibull Dist  | 61.50  | 68.48  | 11.72 | 33.61 |
> | Cars      | Normal Dist   | 51.52  | 58.50  | 18.59 | 32.15 |
> |           | Weibull Dist  | 52.97  | 59.34  | 17.75 | 33.29 |
> | Air       | Normal Dist   | 43.80  | 48.42  | 14.78 | 25.34 |
> |           | Weibull Dist  | 45.24  | 49.74  | 13.46 | 27.29 |
> | Mit       | Normal Dist   | 56.42  | 63.11  | 11.70 | 32.86 |
> |           | Weibull Dist  | 57.10  | 64.15  | 10.30 | 32.87 |
> | Cifar10   | Normal Dist   | 71.13  | 86.18  | 5.60  | 10.95 |
> |           | Weibull Dist  | 73.02  | 87.90  | 3.88  | 11.73 |
> | SVHN      | Normal Dist   | 73.86  | 82.16  | 13.80 | 51.42 |
> |           | Weibull Dist  | 75.21  | 84.32  | 11.64 | 52.01 |
> | Cifar100  | Normal Dist   | 63.51  | 75.82  | 5.64  | 14.25 |
> |           | Weibull Dist  | 64.01  | 76.95  | 4.51  | 14.72 |
> | MNIST     | Normal Dist   | 79.40  | 87.28  | 11.77 | 48.63 |
> |           | Weibull Dist  | 79.90  | 87.82  | 11.23 | 49.15 |
> | Fashion   | Normal Dist   | 59.23  | 64.66  | 28.00 | 27.50 |
> |           | Weibull Dist  | 61.02  | 65.82  | 26.84 | 27.87 |
>
>
>
>
> 4. Thanks for your comment. While achieving 100% accuracy is unrealistic for most machine learning algorithms, our reduction method brings the estimated number of categories much closer to the true value. Users can also fine-tune ζ for their specific dataset to improve accuracy. We opted for a smaller, more general threshold that works well across datasets without tuning, but users can adjust it for even better results when needed.

---

### Review · Reviewer_zvi6 · 2025-11-26

**Summary Of Contributions:**

This manuscript introduces an extreme value theory (EVT)-driven proxy anchoring and reduction framework for continual generalized category discovery. While the proposed approach is novel within this domain and attempts to mitigate over-clustering and cluster drift through EVT-based rejection and proxy reduction mechanisms, the manuscript in its current form suffers from several issues that significantly undermine its clarity, theoretical soundness, and experimental reliability.

**Additional Comments:**

Several writing issues further reduce the readability of the paper. Important variables and symbols are introduced without prior definition in Section 4. A detailed examination of potential errors is required to enhance the accuracy and quality of the paper.

**Audience:**

No

**Audience Explanation:**

The major issue concerns the clarity and organization of the framework. The overall methodology is presented in an overly complex manner, where the main system diagram contains many dense arrows, repeated EVT modules, and several intermediate states whose roles are never explicitly explained in the text. Important components in the figure do not correspond clearly to algorithmic steps or mathematical formulations in the paper, making the methodology unnecessarily difficult to follow and leaving the reader uncertain about the precise flow of operations.

**Claims And Evidence:**

No

**Claims Explanation:**

The theoretical foundation of the EVT-based proxy reduction is insufficiently developed. The paper introduces several functions to define coverage relationships between proxies and then formulates a set-cover-like optimization objective. However, no mathematical derivation is provided to support how EVT-based quantities naturally induce these coverage relations, nor is there an explanation of why the proposed thresholding operation leads to a grounded reduction mechanism.

**Requested Changes:**

1. The manuscript mentions several prior works such as GM, IGCD, MetaGCD, and Proxy Anchor, but it does not provide a meaningful comparative analysis to clarify how the proposed method differs from or improves upon these closely related approaches. The authors are advised to provide further elaboration on this aspect.
2. The experimental evaluation section also contains several weaknesses. Some key claims, such as the effectiveness of the EVT loss or the benefits of proxy reduction, are supported mainly by qualitative visualizations rather than quantitative evidence. Certain performance degradations are attributed to vague “trade-offs” without deeper analysis. Hyperparameters that appear to have significant effect on the results are described only briefly and without sensitivity studies, limiting reproducibility.

---

> ### Author Response · Authors · 2025-12-08
> **Addressed**
>
> We appreciate your thoughtful comments. We have tried to address them the best way we can, as follows:
>
> 1. Thank you for your insightful comment. We have proposed a new approach for handling the CGCD scenario that outperforms previous methods, and we have made an effort to highlight its distinctions throughout the text and figures. We also evaluated our approach on the standard metrics used in the literature and compared it against prior methods across several datasets to demonstrate its advantages. Nonetheless, we can further clarify these distinctions in the revised version.
>
> 2. Thank you for your thoughtful comment. We have presented the effects of the EVT loss and model reduction in our ablation study using multiple combinations, datasets, and metrics. We have also analyzed the impact of all hyperparameters across different components of our method, showing that the model is not highly sensitive to them. Nonetheless, we can provide additional clarification if needed.
>
> 3. We appreciate this comment. We have made every effort to ensure the paper is as clear as possible, introducing and defining variables and symbols in the order they are used. We have also discussed potential sources of error where relevant. However, if there are any specific points that need further clarification, we would be glad to address them.

---

### Decision · Action_Editor_A8ve · 2026-02-03

**Recommendation:** Accept as is

**Audience:**

Yes

**Audience Explanation:**

Given the importance of CGCD, the use of EVT in representation learning and model reduction, and the detailed empirical analysis provided, the work is expected to of interest to at least a subset of the TMLR audience.

**Claims And Evidence:**

Yes

**Claims Explanation:**

This work proposes a unified framework for continual generalized category discovery (CGCD). The core idea of the work is based on integrating extreme value theory (EVT) with proxy-based metric learning. The work introduces an EVT-driven training loss to shape representations, a probability-of-inclusion mechanism for novelty detection, an EVT-based model reduction strategy to mitigate over-clustering during discovery, and a continual learning stage using replay and distillation. The paper is evaluated on multiple standard CGCD benchmarks with extensive experiments and ablations.
The reviews express somewhat differing perspectives on the submission, but also converge on several key points.

Reviewer zvi6 initially had concerns about insufficient theoretical justification for the EVT-based proxy reduction and coverage formulation, lack of clear correspondence between figures, algorithmic steps, and mathematical definitions, and heavy reliance on qualitative, not quantitative, evidence for some claims and limited hyperparameter analysis. The authors responded to these concerns and the manuscript was also revised accordingly.

Reviewer GL8n was largely positive about the methodology and its empirical evaluation. They raised some questions regarding the optimization scheme (e.g., treatment of Weibull parameters) and the choice of clustering algorithms, but these were largely addressed by the authors with clarifications, additional experiments, and revised methodology (e.g., EVT-based replay sampling). Therefore, in my view, these issues do not constitute unresolved gaps between claims and evidence.

Reviewer ZHgS was also largely positive and appreciated the work for its clear problem formulation, well-illustrated and intuitive method, and abundant and convincing evaluation.

Despite some of the reservations of Reviewer zvu6 still persisting, the final discussions acknowledged improvements in clarity, logical structure, variable definitions, figures, and supplementary experiments, including clustering comparisons and the Weibull-based feature sampling strategy. While there were still some concerns about the overall direction of the work, they are more about perceived conceptual limitations, rather than gaps between stated claims and experimental evidence.

In my opinion, while not all reviewers fully endorse the theoretical depth of every component, the revised manuscript provides sufficient empirical evidence and clarification to support its claims.